# Deep density networks and uncertainty in recommender systems

## Abstract

Building robust online content recommendation systems requires learning complex interactions between user preferences and content features. The field has evolved rapidly in recent years from traditional multi-arm bandit and collaborative filtering techniques, with new methods integrating Deep Learning models that enable to capture non-linear feature interactions. Despite progress, the dynamic nature of online recommendations still poses great challenges, such as finding the delicate balance between exploration and exploitation. In this paper we provide a novel method, Deep Density Networks (DDN) which deconvolves measurement and data uncertainties and predicts probability density of CTR (Click Through Rate), enabling us to perform more efficient exploration of the feature space. We show the usefulness of using DDN online in a real world content recommendation system that serves billions of recommendations per day, and present online and offline results to evaluate the benefit of using DDN.

## 1 Introduction

In order to navigate the vast amounts of content on the internet, users either rely on active search queries, or on passive content recommendations. As the amount of the content on the internet grows, content discovery becomes an increasingly crucial challenge, shaping the way content is consumed by users. Taboola's content discovery platform aims to perform *"reverse search"*, using computational models to match content to users who are likely to engage with it. Taboola's content recommendations are shown in widgets that are usually placed at the bottom of articles (see Fig. 1) in various websites across the internet, and serve billions of recommendation per day, with a user base of hundreds of millions of active users.

Traditionally recommender systems have been modeled in a multi-arm bandit setting, in which the goal is to a find a strategy that balances *exploitation* and *exploration* in order to maximize the long term reward. Exploitation regimes try to maximize the immediate reward given the available information, while exploration seeks to extract new information from the feature space, subsequently increasing the performance of the exploitation module.

One of the simplest approaches to deal with multi-arm bandit problems is the $\epsilon$-greedy algorithm, in which with probability $\epsilon$ a random recommendation is chosen, and with probability $1 - \epsilon$ the recommendation with the highest predicted reward is chosen. Upper Confidence Bound -UCB- (Auer et al. (2002)) and Thompson sampling techniques (Thompson (1933)) use prediction uncertainty estimations in order to perform more efficient exploration of the feature space, either by explicitly adding the uncertainty to the estimation (UCB) or by sampling from the posterior distribution (Thompson sampling). Estimating prediction uncertainty is crucial in order to utilize these methods. Online recommendations are noisy and probabilistic by nature, with measured values being only a proxy to the true underlying distribution, leading to additional interesting challenges when predicting uncertainty estimations.

In this paper we present DDN, a unified deep neural network model which incorporates both measurement and data uncertainty, having the ability to be trained end-to-end while facilitating the exploitation/exploration selection strategy. We introduce a mathematical formulation to deconvolve measurement noise, and to provide data uncertainty predictions that can be utilized to improve exploration methods. Finally, we demonstrate the benefit of using DDN in a real world content recommendation system.

Figure 1: Taboola's recommendation widget example.

## 2 RELATED WORK

Over the past decade deep learning has been applied with tremendous success in many different application domains such as computer vision, speech recognition and machine translation. In recent years we have seen a corresponding explosion of deep learning models in the recommender systems landscape, revolutionizing recommendation architectures and providing superior performance over traditional models ( Zhang et al. (2017); Cheng et al. (2016); Covington et al. (2016); Okura et al. (2017)). Deep learning's ability to capture non-linearities has enabled to model complex user-item relations and to integrate higher level representations of data sources such as contextual, textual and visual input.

Traditionally recommender systems have been modeled in a multi-arm bandit setting, where the goal is to find an exploitation/exploration selection strategy in order to maximize the long term reward. A similar challenge has been faced in Reinforcement learning (RL) setting, in which an agent has to decide when to forego an immediate reward and to explore its environment. Bayesian neural networks (Neal (2012)) using distributions over the weights were applied by using either sampling or stochastic variational inference (Kingma & Welling (2013); Rezende et al. (2014)). While Bayesian models offer a mathematically grounded framework, they usually entail a prohibitive computational cost. Blundell et al. (2015) proposed Bayes by Backprop algorithm for the variational posterior estimation and applied Thompson sampling. Gal & Ghahramani (2016) proposed Monte Carlo (MC) dropout, a Bayesian approximation of model uncertainty by extracting estimations from the different sub-models that have been trained using dropout. Kendall & Gal (2017) separated uncertainty into two types, model and data uncertainty, while studying the effect of each uncertainty separately in computer vision tasks. Li et al. (2010a) formulated the exploration/exploitation trade-off in personalized article recommendation as a contextual bandit problem and proposed LinUCB algorithm, which adapts the UCB strategy to support models based on contextual features.

The effect of measurement noise and noisy labels has been studied extensively (Frénay & Verleysen (2014)). Mnih & Hinton (2012) proposed a probabilistic model for the conditional probability of seeing a wrong label, where the correct label is a latent variable of the model. Goldberger & Ben-Reuven (2017) explicitly modelled noise by an additional softmax layer that connects the correct labels to the noisy ones. In this paper we model measurement noise using a Gaussian model and combine it with a MDN.

## 3 TABOOLA'S RECOMMENDER SYSTEM OVERVIEW

Taboola's revenue stream is facilitated by online advertisers, who pay a fixed amount *CPC* (*Cost Per Click*) for each user that is redirected to their site after clicking on a Taboola's recommendation. The algorithm's total value is measured in *RPM* (*Revenue Per Mille*) where $RPM = CTR * CPC * 1000$, is the average revenue for every 1000 recommendations and *CTR* (*Click Through Rate*) is the probability of a recommendation being clicked. Content recommendations are ranked according to

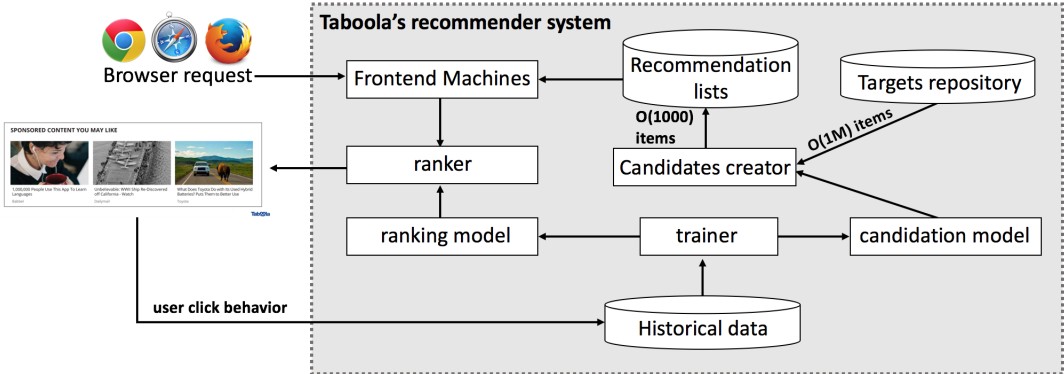

Figure 2: High level overview of candidation and ranking architecture.

their predicted RPM; recommendations with the highest predicted RPM will be shown to the user. Taboola's main algorithmic challenge is to provide an estimate of the CTR in any given context.

Taboola's recommendation engine needs to provide recommendations within strict time constraints ($< 50ms$). It is infeasable to rank millions of recommendations in that time frame; in order to support this we have partitioned the system into a two-step process, *candidation* and *ranking* (see Fig. 2). During the candidation phase, we narrow down the list of possible recommendations to thousands based on their RPM prediction in a specific context. CTR prediction in this setting is based on features such as the creative of recommendations (text and image) and empirical click statistics. This relatively small list of recommendations is written to distributed databases in worldwide data centers, and are re-calculated by Taboola's servers continuously throughout the day. When the frontend servers get a request for recommendations from the browser, they retrieve the relevant ready-made recommendation list, and perform an additional ranking of the recommendations based on additional user features using a DNN, further personalizing recommendations. This system architecture shows similarities to (Cheng et al. (2016)).

The dynamic nature of Taboola's marketplace means that our algorithm constantly needs to evaluate new recommendations, with tens of thousands of new possible recommendations every day. To support this, we split the algorithm into *exploration* and *exploitation* modules. The exploitation module aims to choose the recommendations that maximize the RPM, while the exploration module aims to enrich the dataset available for exploitation models by showing new recommendations.

In this paper we focus on the candidation phase and the corresponding CTR prediction task, leaving out of this scope the second ranking step.

## 4 DEEP DENSITY NETWORK AND UNCERTAINTY

In this section we present Deep Density Network (DDN) and describe its ability to deconvolve measurement noise and integrate it with data uncertainty in a unified model. Employing uncertainty during the training phase can be interpreted as loss attenuation, making our model more robust to noisy data. In addition, accurate uncertainty estimations enable us to employ more efficient exploitation/exploration selection strategies as discussed below.

### 4.1 DEEP ARCHITECTURE

Our deep recommender model is a hybrid of a content-based and a collaborative filtering (CF) recommendation system. A high level overview is depicted in Fig. 3. We use two separate subnets which model the target and the context features. The target subnet receives as input the content features seen by the user and additional features such as the recommendation age which are unseen to the user. The categorical features are passed through an embedding layer and concatenated along with the numerical features, followed by fully connected layers with a RELU activation function. The result is the target feature descriptor. Similarly, the context features are modeled using a separate

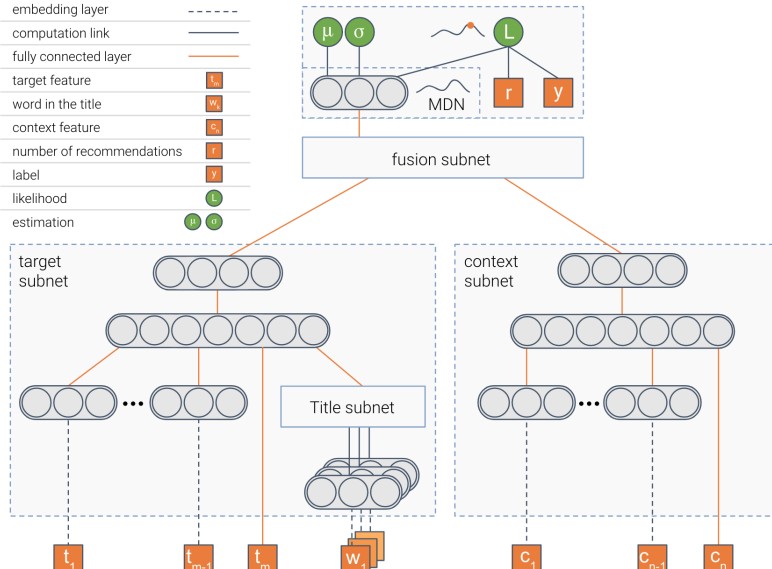

Figure 3: A hybrid content-based and collaborative filtering model accounting for data and measurement uncertainties. Both target and context features are passed through a DNN, then through a fusion sub-network, and finally through a MDN layer that outputs the parameters of a distribution over CTR. The number of recommendations $r$ is used for attenuating the loss of noisy examples.

DNN, taking as input context features such as device type where the target will be recommended, resulting in the context feature descriptor. The target and context feature descriptors are then fused using a DNN which outputs both the CTR and its uncertainty prediction, with the measurement being compensated as described in sec. 4.2.

In order to train our models, we collect and use historical data which consists of target and context pairs $(t, c)$, where $t$ is the target we recommended in a specific browsing context $c$. Each row in our training dataset includes the empirical CTR of the target-context pair $(t, c)$, together with additional target and context features. We train models that optimize CTR prediction on this dataset using Maximum Likelihood Estimation (MLE) as discussed below.

## 4.2 MODELING UNCERTAINTY

We separate uncertainty into two different types: measurement and data uncertainty. Measurement uncertainty corresponds to the uncertainty of our observation due to the measurement noise introduced by the binomial recommendation experiment. This type of uncertainty depends on the number of times $r$, a specific target $x = (t, c)$ pair was recommended, i.e. target $t$ was recommended in context $c$. Data uncertainty corresponds to the inherent noise of the observations; In contrast to measurement noise, it cannot be reduced even if more data was to be collected, as it corresponds to the inherent variability of the data. Data uncertainty is categorized into homoscedastic and heteroscedastic uncertainty. Homoscedastic is constant over all different inputs, in contrast to heteroscedastic which depends on the inputs, i.e. different input values may have more noisy outputs than others.

Simple algorithms like $\epsilon$-greedy choose actions indiscriminately during exploration, with no specific preference for targets that have higher probability to be successful in exploitation, or for targets that hold significant information gain about the feature space, for example targets that contain words that weren't previously recommended. It is beneficial to select among the non-greedy actions according to their potential of actually being optimal, taking into account both the expectation and the variance of their CTR estimates. Estimating uncertainty enables us to employ the upper confidence bound (UCB) algorithm for a better and adaptive selection strategy between exploitation/exploration.

We estimate both the mean payoff $\mu^t$ and the standard deviation $\sigma^t$ of each target $t$ and select the target that achieves the highest UCB score, where $a$ is a tunable parameter:

$$A = \arg\max_t(\mu^t + a \cdot \sigma^t) \tag{1}$$

Our marketplace is defined by a very high recommendation turnover rate, with new content being uploaded everyday and old one becoming obsolete. Probabilistic modeling of the data uncertainty assists us in using the exploration model in order to sample targets that have the highest potential value, by employing the UCB strategy.

In contrast to the variance captured by data uncertainty, model uncertainty corresponds to what the model "knows" about the feature space. Gal & Ghahramani (2016) show that model uncertainty can capture the confidence about different values in the feature space. This however comes at a prohibitive computational cost when calculated using dropout. We explore the feature space by setting to Out Of Vocabulary (OOV) categorical feature values which have been shown less than a minimal threshold. As shown in Fig.4, OOV values indeed get larger uncertainty estimations.

### 4.2.1 INTEGRATING MEASUREMENT NOISE

In order to deconvolve the data and measurement uncertainties we explicitly model them together. Let $Y$, $Y^*$ and $\epsilon$ be three random variables given $x = (t, c)$. $Y$ corresponds to observed CTR, after recommending $(t, c)$ pair, $r$ times. $Y^*$ corresponds to the true/clean CTR without the measurement noise, i.e. the observed CTR had we recommended $t$ infinite times in $c$. $\epsilon$ corresponds to the binomial noise error distribution.

$$Y = Y^* + \epsilon \tag{2}$$

We are modelling data uncertainty by placing a distribution over the output of the model and learning it as a function of the different inputs. To this end, we are using Mixture Density Network (MDN), which employ a Gaussian Mixture Model (GMM) to model $Y^*$ (Bishop (1994)).

$$Y^* \sim \sum_i \alpha_i \mathcal{N}(\mu_i, \sigma_i^2) \tag{3}$$

For every input the MDN model predicts the coefficients of the GMM; These are the mixing coefficients, $\alpha_i$, $\mu_i$ and $\sigma_i$, from which we estimate the expected value and the standard deviation of $Y^*$.

The measurement uncertainty $\epsilon$ corresponds to the measurement noise distribution which we approximate with a Gaussian distribution:

$$\epsilon \sim \mathcal{N}(0, \sigma_\epsilon^2) \tag{4}$$

Due to the fact that data noise is small given $x$, we enforce constant $\sigma_\epsilon = f(\mu, r)$ for every $y^*|x$ where $\mu$ is the expected value of $y^*|x$. In this way, $Y^* \perp\!\!\!\perp \epsilon$ given $x$, as $\sigma_\epsilon$ depends only on $r$ and $\mu$. We can rewrite eq. 2 using eq. 3 and 6 to:

$$Y \sim \sum_i \alpha_i \mathcal{N}(\mu_i, \sigma_i^2 + \sigma_\epsilon^2) \tag{5}$$

This enables us to deconvolve and model both data and measurement uncertainties, using a single model which combines MDN and a Gaussian model. Given this probability estimation, the training process uses SGD for minimizing the loss:

$$L = -\log(P(y|x)) \tag{6}$$

## 5 EXPERIMENTS

### 5.1 DATA AND MODELS

*Data:* For the purpose of this paper we use the browsed website (i.e. *publisher*) as the user context. In all of the experiments we used three months of historical data, containing $\sim$10M records of

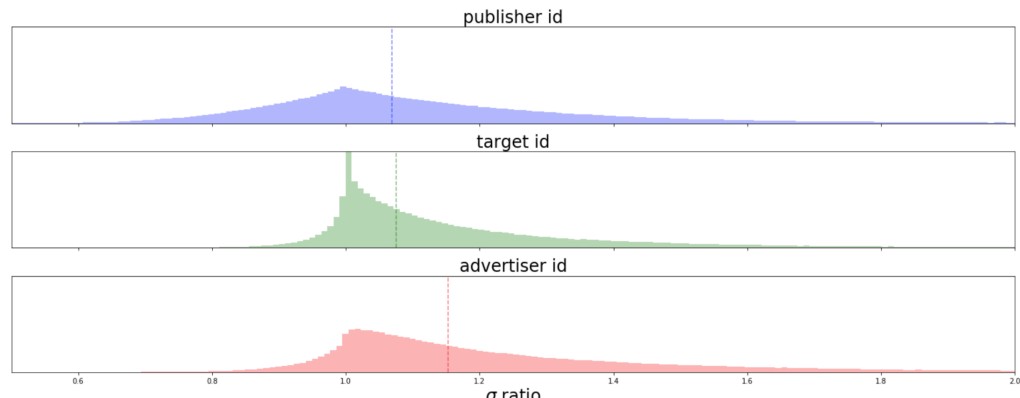

Figure 4: Ratio of estimated $\sigma$, before and after feature values are set to OOV. Dashed vertical lines are the median values.

| Dataset | MDN | DDN | Improvement |
|---------|---------|---------|-------------|
| D1 | 0.2681 | 0.25368 | 5.3% |
| D2 | 0.25046 | 0.24367 | 2.7 % |

Table 1: Relative improvement in the MSE between MDN and DDN when trained over two datasets that differ by the amount of measurement noise.

target-publisher pairs. The dataset contains $\sim$ 1M unique targets and $\sim$10K unique publishers. Every experiment has been run on multiple time slots to validate that the results were statistically significant.

*Models:* We have experimented with the following models:

1. **REG** is a regression model that outputs a point estimate of the CTR where the loss is the MSE between the actual and the predicted CTR.

2. **MDN** is a model that estimates the distribution over the CTR utilizing a mixture of Gaussians (see sec. 4.2.1).

3. **DDN** is a model that estimates the distribution over the CTR combining the data and measurement uncertainties (see sec. 4.2.1).

In order to have a fair comparison, we tuned the hyper-parameters (e.g. embedding sizes, number of layers, number of mixtures) for each model separately; we performed thousands of iterations of random search, and chose the parameters that yielded the best results.

## 5.2 METRICS AND EVALUATION

We evaluate our models using Mean Square Error (MSE). Due to the dynamic nature of online recommendations it is crucial that we evaluate our models online within an A/B testing framework, by measuring the average RPM of models across different publishers. In addition we utilize an online throughput metric which aims to capture the effectiveness of the exploration mechanism. Prior works have put an effort on how to measure exploration; Li et al. (2010b) built an offline simulator that enabled them to test different models to see which one achieves target coverage faster. This is not feasible in our case given the large turnover rate in the recommendation pool. Instead, we use the following targets throughput metric.

Let $< t_i, p_i >$ be the set of target-publisher pairs that accumulated enough data to achieve empiric binomial statistical significance in a given day. A model is said to be contributing to $< t_i, p_i >$ if it has recommended $t_i$ in $p_i$ in the previous day more times than a predefined threshold. Our throughput metric is defined by the number of targets that a specific model contributed to this set.

| Model | REG | MDN | DDN |
|---|---|---|---|
| RPM lift | 0% | 1.2% | 2.9% |

Table 2: A comparison of the online RPM lift between the different models.

| $a$ | 0 | 0.5 | 1 | 1.5 |
|---|---|---|---|---|
| RPM lift | 0% | -0.05% | -0.2% | -0.3% |
| Throughput lift | 0% | 6.5% | 9.1% | 11.7% |

Table 3: RPM lift vs targets throughput as a function of different values of $c$.

## 5.3 EXPERIMENTAL RESULTS

**Feature importance:** Understanding the parameters of deep learning networks poses a significant challenge compared to linear and tree based models. We utilize the fact that our models output a full distribution rather than a point estimate to evaluate feature importance. In our analysis, we evaluate the effect on the $\sigma$ prediction when a feature is "hidden" from the model during inference, by setting it to OOV. For each feature, we calculate statistics over the ratio $\sigma_{oov}/\sigma$, between the predicted $\sigma$ before and after setting it to OOV.

In Fig. 4 we observe that the analyzed features have a large impact on data uncertainty. The median values of the various features are greater than one, validating our assumption that feature values that did not appear in the training data will obtain a higher uncertainty estimation. In addition, we see a distinct ordering of feature importance, where new advertisers yield a larger ratio than new targets. Using $\sigma$ in a UCB setting (as in equation 1) will prioritize new targets, especially ones from new advertisers - a desired behaviour both in terms of information gain and advertiser satisfaction.

**Measurement noise:** In Table 1 we compare the MDN and DDN models by training them on two different datasets, D1 and D2. D1 differs from D2 by the amount of noise in the training samples; D1 contains noisy data points with relatively small amount of empirical data, while D2 contains examples with higher empirical statistical significance. We observe that DDN improves on MDN performance by 2.7% when using D1 for training, and by 5.3% when using D2. This validates that integrating measurement noise into our modeling is crucial when the training data contains very noisy samples, by attenuating the impact of measurement noise on the loss function. (see sec. 4.2.1)

**Model comparison:** In Table 2 we compare the three different models discussed previously in terms of online RPM. We observe that DDN is the best performing model, outperforming MDN and REG by 1.7% and 2.9% respectively. These results verify once again that the loss attenuation achieved by DDN during training has enabled it to converge to better parameters, generalizing better to unseen examples.

**RPM lift vs. targets throughput:** We analyzed the effect of the parameter $a$ found in 1. From a theoretical standpoint, increasing this value is supposed to prioritize higher information gain at the expense of RPM, by choosing targets that the model is uncertain about. This trade-off is worthwhile in the long term. In Table 3 we observe that there is an inverse correlation between RPM and throughput which is triggered by different values of $a$, with targets throughput increasing by 11.7% when setting $a = 1.5$. Choosing the right trade-off is an application specific concern, and we chose the trade-off induced by $a = 0.5$, resulting in a good throughput gain with a small RPM cost.

## 6 CONCLUSIONS

We have introduced Deep Density Network (DDN), a unified DNN model that is able to predict probability distributions and to deconvolve measurement and data uncertainties. DDN is able to model non-linearities and capture complex target-context relations, incorporating higher level representations of data sources such as contextual and textual input. We have shown the added value of using DNN in a multi-arm bandit setting, yielding an adaptive selection strategy that balances *exploitation* and *exploration* and maximizes the long term reward. We presented results validating DDN's improved noise handling capabilities, leading to 5.3% improvement on a noisy dataset.

Furthermore, we observed that DDN outperformed both REG and MDN models in online experiments, leading to RPM improvements of 2.9% and 1.7% respectively. Finally, by employing DDN's data uncertainty estimation and UCB strategy, we improved our exploration strategy, depicting 6.5% increase of targets throughput with only 0.05% RPM decrease.

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
