# OpenReview forum: "DEEP DENSITY NETWORKS AND UNCERTAINTY IN RECOMMENDER SYSTEMS"
_ICLR.cc/2018/Conference — Reject_

### Official Review · AnonReviewer1 · 2017-11-21
**Interesting scalable idea but needs more support**

**Rating:** 4
**Confidence:** 5

**Review:**

The paper adresses a very interesting question about the handling of the dynamics of a recommender systems at scale (here for linking to some articles).
The defended idea is to use the context to fit a mixture of Gaussian with a NN and to assume that the noise could be additively split into two terms. One depend only on the number of observations of the given context and the average reward in this situation and the second term begin the noise. This is equivalent to separate a local estimation error from the noise.

The idea is interesting but maybe not pushed far enough in the paper:
*At fixed context x, assuming that the error is a function of the average reward u and of the number of displays r of the context could be a constant could be a little bit more supported (this is a variance explanation that could be tested statistically, or the shape of this 2D function f(u,r) could be plot to exhibit its regularity).
* None of the experiments is done on public data which lead to an impossible to reproduce paper
* The proposed baselines are not really the state of the art (Factorization Machines, GBDT features,...) and the used loss is MSE which is strange in the context of CTR prediction (logistic loss would be a more natural choice)
* I'm not confident with the proposed surrogate metrics. In the paper, the work of Lihong Li &al on offline evaluation on contextual bandits is mentioned and considered as infeasible here because of the renewal of the set of recommendation. Actually this work can be adapted to handle theses situations (possibly requiring to bootstrap if the set is actually regenerating too fast). Also note that Yahoo Research R6a - R6b  datasets where used in ICML'12 Exploration and Exploitation 3 challenge where about pushing some news in a given context and could be reused to support the proposed approach. An other option would be to use some counterfactual estimates (See Leon Bottou &all and Thorsten Joachims &all)
* If the claim is about a better exploration,  I'd like to have an idea of the influence of the tuning parameters and possibly a discussion/comparison over alternatives strategies (including an epsilon-n greedy algorithm)

Besides theses core concerns, the papers suffers of some imprecisions on the notations which should be clarified.
* As an example using O(1000) and O(1M) in the figure one. Everyone understands what is meant but O notation are made to eliminate constant terms and O(1) = O(1000).
* For eqn (1) it would be better to refer to and "optimistic strategy" rather to UCB because the name is already taken by an algorithm which is not this one. Moreover the given strategy would achieve a linear regret if used as described in the paper which is not desirable for bandits algorithms (smallest counter example with two arms following a Bernouilli with different parameters if the best arms generates two zero in a row at the beginning, it is now stuck with a zero mean and zero variance estimate). This is why bandits bounds include a term which increase with the total number of plays. I agree that in practice this effect can be mitigated at that the strategy can be correct in the contextual case (but then I'd like to the dependancies on x to be clear)
* The papers never mentions whats is a scalar, a vector or a matrix. This creates confusion: as an example eqn (3) can have several different meaning depending if the values are scalars, scalars depending on x or having a diagonal \sigma matrix
* In the paragraph above (2) I unsure of what is a "binomial noise error distribution" for epsilon, but a few lines later epsilon becomes a gaussian why not just mention that you assume the presence of a gaussian noise on the parameters of a Bernouilli distribution ?

---

### Official Review · AnonReviewer3 · 2017-11-27
**Unclear contribution, questionnable approach**

**Rating:** 3
**Confidence:** 4

**Review:**

In the paper "DEEP DENSITY NETWORKS AND UNCERTAINTY IN RECOMMENDER SYSTEMS", the authors propose a novel neural architecture for online recommendation. The proposed model deals with data and measurement uncertaintes to define exploitation/exploration startegies.

My main concern with the paper is that the contribution is unclear, as the authors failed from my point of view in establishing the novely w.r.t. the state of the art regarding uncertainty in neural networks. The state of the art section is very confusing, with works given in a random order, without any clear explanation about the limits of the existing works in the context of the task addressed in the paper. The only positioning argument that is given in that section is the final sentence "In this paper we model measurement noise using a Gaussian model and combine it with a MDN". It is clearly not sufficient to me, as it does not gives insights about why such proposal is done.

In the same spirit, I cannot understand why not any classical bandit baseline is given in the experiments. The experiments only concern two slightly different versions of the proposed algorithm in order to show the importance of the deconvolution of both considered noises, but nothing indicates that the model performs fairly well compared to existing approaches. Also, it would have been useful to compare ot to other neural models dealing with uncertainty (some of them having been applied to bandit problems- e.g., Blundell et al. (2015)).

At last, for me the uncertainty considered in the proposal is not sufficient to claim that the approach is an UCB-like one. The confidence bound considered should include the uncertainty on the parameters in the predictive posterior reward distribution (as done for instance in Blundell et al. (2015) in the context of neural networks), not only the distribution of the observed data with regards to the considered probabilistic families. Not enough discussion is given wrt the assumptions made by the model anyway. The section 4 is also particularly hard to follow.

Other remarks:
    - Equation (1) does not fit with mixtures considered in 4.2.1. So what is the selection score that is used
    - "Due to the fact that data noise is small given x" => what does it mean since x is a couple ? Also I cannot understand the causal relation with the following of the sentence
    - Figure 4 (and the associated paragraph) is very difficult to understand (I couldn't extract any information from this)
    - Too many abreviations that complicate the reading
    - The throughput measure is not clear
    - Not enough justification about the architecture. For instance, nothing is said about the title subnet represented in the figure 3.
    - What is the "recommendation age" ?
    - "We can rewrite eq. 2 using eq. 3 and 6" => "and 4".

---

### Official Review · AnonReviewer2 · 2017-11-30
**A paper that needs some rewriting to be more clear to judge its quality**

**Rating:** 4
**Confidence:** 3

**Review:**

This paper presents a methodology to allow us to be able to measure uncertainty of the deep neural network predictions, and then apply explore-exploit algorithms such as UCB to obtain better performance in online content recommendation systems. The method presented in this paper seems to be novel but lacks clarity unfortunately. My main doubt comes from Section 4.2.1, as I am not sure how exactly the two subnets fed into MDN to produce both mean and variance, through another gaussian mixture model. More specifically, I am not able to see how the output of the two subnets get used in the Gaussian mixture model, and also how the variance of the prediction is determined here. Some rewriting is needed there to make this paper better understandable in my opinion.

My other concerns of this paper include:
1. It looks like the training data uses empirical CTR of (t,c) as ground truth. This doesn't look realistic at all, as most of the time (t,c) pair either has no data or very little data in the real world. Otherwise it is a very simple problem to solve, as you can just simply assume it's a independent binomial model for each (t,c).
2. In Section 4.2.1, CTR is modeled as a Gaussian mixture, which doesn't look quite right, as CTR is between (0,1).
3. A detailed explanation of the difference between MDN and DDN is needed.
4. What is OOV in Section 5.3?

---

### Decision · Program_Chairs · 2018-01-29
**ICLR 2018 Conference Acceptance Decision**

**Decision:**

Reject

**Comment:**

Meta score: 4

The paper concerns the development of a density network for estimating uncertainty in recommender systems.  The submitted paper is not very clear and it is hard to completely understand the proposed method from the way it is presented.  This makes assessing the contribution of the paper  difficult.

Pros:
 - addresses an  interesting and important problem
 - possible novel contribution

Cons:
 - poorly written, hard to understand precisely what is done
 - difficult to compare with the state-of-the-art, not helped by disorganised literature review
 - experimentation could be improved

The paper needs more work before being ready for publication.